# ECG Signal Denoising Method Based on Disentangled Autoencoder

**Haicai Lin** [1,2], **Ruixia Liu** [2,*] and **Zhaoyang Liu** [2]

1.  School of Mathematics and Statistics, Qilu University of Technology (Shandong Academy of Sciences), Jinan 250353, China
2.  Shandong Artificial Intelligence Institute, Qilu University of Technology (Shandong Academy of Sciences), Jinan 250014, China
*   Correspondence: liurx@sdas.org

**Abstract:** The electrocardiogram (ECG) is widely used in medicine because it can provide basic information about different types of heart disease. However, ECG data are usually disturbed by various types of noise, which can lead to errors in diagnosis by doctors. To address this problem, this study proposes a method for denoising ECG based on disentangled autoencoders. A disentangled autoencoder is an improved autoencoder suitable for denoising ECG data. In our proposed method, we use a disentangled autoencoder model based on a fully convolutional neural network to effectively separate the clean ECG data from the noise. Unlike conventional autoencoders, we disentangle the features of the coding hidden layer to separate the signal-coding features from the noise-coding features. We performed simulation experiments on the MIT-BIH Arrhythmia Database and found that the algorithm had better noise reduction results when dealing with four different types of noise. In particular, using our method, the average improved signal-to-noise ratios for the three noises in the MIT-BIH Noise Stress Test Database were 27.45 db for baseline wander, 25.72 db for muscle artefacts, and 29.91 db for electrode motion artefacts. Compared to a denoising autoencoder based on a fully convolutional neural network (FCN), the signal-to-noise ratio was improved by an average of 12.57%. We can conclude that the model has scientific validity. At the same time, our noise reduction method can effectively remove noise while preserving the important information conveyed by the original signal.

**Keywords:** disentangled representation learning; autoencoder; ECG signal denoising; deep learning

## 1. Introduction

At present, cardiovascular disease is one of the major threats to human life and health, and the number of deaths due to cardiovascular disease is increasing every year [1]. The electrocardiogram (ECG) is an important tool for cardiology research, and it is also a powerful basis for doctors to directly analyze the cardiac status of patients. Compared with other methods, ECG is often highly efficient and non-invasive and has low costs [2]. As a bioelectric signal source, the signal intensity of the heart must be directly related to the number of active cells, and the number of heart cells constituting the atrium and ventricle is the largest. Therefore, a surface ECG waveform mainly reflects changes in the action potentials of the atrial and ventricular cells [3]. Figure 1 shows a complete ECG cycle in which the P wave, QRS bundle, and T wave are the most important characteristic waves. These waves and the PR interval, QT interval, and ST segment formed on their basis are the most important characteristic information of the ECG [4] and can reflect the conduction system of the heart and whether the heart itself has lesions from many aspects. Therefore, in the process of collecting ECG data, it is particularly important to ensure that the ECG is not disturbed by noise.

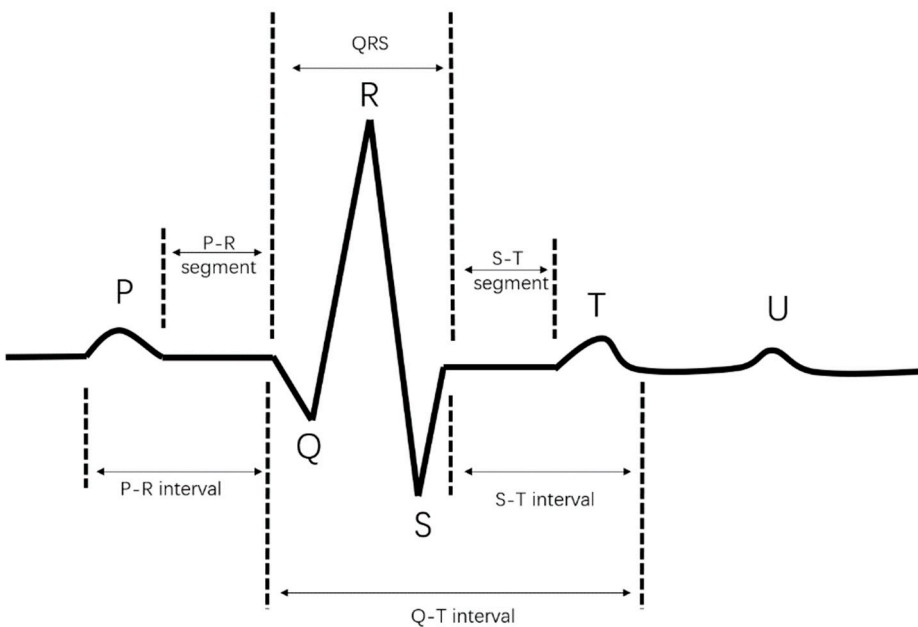

**Figure 1.** A complete ECG cycle.

The ECG has the characteristics of low frequency and energy concentration, and the signal is weak, easily disturbed by noise, and also has quasi-periodicity [5]. ECG sampling is often accompanied by a lot of noise, mainly muscle artefacts (MA) [6], electrode motion artefacts (EM) [7], and baseline wander (BW) [8]. Muscle artefacts are a common type of high-frequency noise, usually between 30 and 300 Hz. The source of this noise comes from the tremor of the body's muscles and therefore appears on the image as a discrepancy between the muscle image and the actual situation. In order to reduce the impact of this noise on the image quality, medical imaging professionals must use appropriate techniques to reduce its effects. Electrode motion artefacts are instantaneous noise caused by poor contact between the skin and the electrodes. Baseline wander is low-frequency noise, mainly caused by breathing, and electrode slippage is caused by low-frequency interference. These noises have a major impact on a clinician's ability to diagnose the nature of a disease, which is likely to lead to a false diagnosis. Therefore, it is particularly important to denoise sampled ECG data. The purpose of this paper is to effectively reduce the noise in ECG data in order to improve the signal-to-noise ratio of the signal, accurately detect ECG data, improve the visualization and analysis of ECG data, and effectively improve the accuracy of diagnosis. We have been involved in researching a noise reduction method that separates the depth characteristics of ECG data from noisy signals, with the aim of reducing unnecessary noise, improving the accuracy of ECG data, better detecting heart disease, and improving healthcare.

## 2. Related Work

### 2.1. The Traditional ECG Denoising Method

The ECG acquisition process is often accompanied by a large amount of noise, which seriously affects a doctor's accurate diagnosis of a patient. Therefore, technical research on removing noise from ECG data has always been a hot topic. Many researchers have proposed different research algorithms. In 2015, the authors of [9] proposed the use of multivariate empirical mode decomposition (MEMD) to remove baseline wander in ECG data. MEMD technology was a multivariate extension of EMD, which had recently attracted the attention of researchers for many applications [10]. The basic idea of this method was to transform a signal into a multi-channel signal and use the MEMD algorithm to process the multi-channel signal at the same time. The last intrinsic mode function (IMF) or the last two IMFs obtained by decomposition were removed from the ECG as

the baseline wander to obtain a baseline-corrected ECG. Simulation results showed that this method can remove the baseline wander in ECG data and retain the morphological characteristics of ECG data. Empirical mode decomposition technology was a data-driven adaptive threshold method that was very suitable for non-stationary signals such as ECG data [11]. However, it could not select IMF efficiently and adaptively, which led to the loss of information. The authors of [12] proposed a noise removal method based on empirical mode decomposition combined with a wavelet threshold for adaptive ECG baseline wander. ECG data containing BW noise were denoised. First, they were decomposed into high-frequency signals and low-frequency signals using empirical mode decomposition. Then, the low-frequency signals were wavelet-transformed, and the high-frequency signals were combined to reconstruct the ECG data, thereby achieving the effect of removing baseline wander noise. The technology based on the wavelet transform was also more popular and widely used because of its ability to characterize the time–frequency domain information of time-domain signals [13]. The noise reduction method based on wavelets was widely used because it had good localization properties and could fully highlight the detailed features of ECG data in the time domain and frequency domain. However, due to the selection of the wavelet basis function and threshold, the amplitudes of the R wave and S wave may have been reduced after denoising [14]. Therefore, how to select the appropriate wavelet basis function and wavelet decomposition level to remove the noise in the input signal was still a problem [15]. In the wavelet denoising method, two thresholds [16] were used to enhance the ECG data. Tan Xue et al. [17] proposed an improved wavelet-threshold-function denoising method that avoided the defect of poor continuity at the threshold after processing the signal in the traditional soft and hard threshold function. Das et al. [18] compared the wavelet transform and proposed an ECG denoising method based on the S-transform. Because of the sparse characteristics of the ECG itself [19], a method based on sparse representation had been studied a lot in recent years. However, some ladder components were introduced after sparse noise reduction [20], which made the signal uneven after noise reduction, and most sparse representations use the L1 norm as a penalty term, which leads to the underestimation of the original signal [21].

In the field of deep learning, an ECG denoising method based on the autoencoder model [22] was more popular. An autoencoder composed of eight convolution blocks and eight deconvolution blocks was proposed by Eleni et al. [23], which can effectively learn the characteristics of ECGs and remove noise. As one of the many breakthroughs in deep learning technology, a generative adversarial network (GAN) had been widely used. Since Goodfellow [24] first introduced this method, many variants of generative adversarial networks have been developed. Radford et al. [25] proposed a deep convolutional generative adversarial network (DCGAN) in 2015. The DCGAN used a convolutional layer to replace the fully connected layer and replaced the original pooling layer with a convolution of the same step size. Pratik Singh and Gayahar Pradhan [26] proposed a generative adversarial network architecture for ECG denoising. The GAN model based on convolutional neural networks was effectively trained for ECG noise filtering, and end-to-end GAN model training was performed using clean and noisy ECG data. Zhu et al. [27] proposed a generative adversarial network, which was a Bi-LSTM-CNN network composed of bidirectional long short-term memory (LSTM) and a convolutional neural network (CNN). The model included a generator and a discriminator. The generator used a two-layer Bi-LSTM network, and the discriminator was based on a convolutional neural network.

### 2.2. Disentangled Representation Learning

Disentangled representation learning [28] was first proposed by Bengio in 2013. The feature sets being trained may be used for multiple tasks, which may have different relevant feature subsets. Therefore, the most robust method of feature learning was to separate as many factors as possible and discard as little information as possible. The application of disentangled representation learning makes the black box project of deep learning more interpretable and can show the specific meaning and function of each layer of the neural

network. The traditional denoising autoencoder maps the signal information and noise information to the hidden space together in the encoding phase and then decodes them into signals through the hidden space variable. We believe that there is a phenomenon of decoding confusion; that is, some noise information is decoded into signal information together, thus reducing the noise reduction effect of the signal. Therefore, we are trying to decouple the noise information and signal information in the hidden space, eliminate the entanglement of signal information and noise information, and then decode the clean signal.

### 2.3. The Major Contributions of the Paper

Although the ECG denoising method based on a denoising autoencoder is robust to different types of noise, it is still a problem that the signal features and noise features are not completely separated, resulting in the underestimation of some ECG details. Based on this, our contributions are as follows:

(1) Based on the denoising autoencoder, we introduce the disentangled mechanism and propose a new disentangled autoencoder network (DANet) for ECG denoising processing, which solves the problem of the incomplete separation of signal features and noise features.
(2) In the realization of the disentangled mechanism, we use attention to shunt latent variables, deconstruct the potential spatial separation between noise and the original ECG data, and finally use two decoders to approximate the original signal and noise.
(3) Experiments demonstrated that our proposed method achieved optimal performance and can effectively preserve useful detail characteristics.

The overall structure of this paper is as follows: Section 2 provides the related work. Section 3 focuses on the details of the database we used and how the data were processed. Section 4 details the implementation of the algorithm, the coding of ECG features, the separation of signal and noise characteristics, and the signal recovery process. Section 5 presents experiments comparing the denoising effects on ECG data and four different types of noise. Sections 6 and 7 provide a general discussion and a summary of our proposed method.

## 3. The Materials

### 3.1. Dataset Description

We used the MIT-BIH Noise Stress Test Database and the Arrhythmia Database as the main dataset [29]. The MIT-BIH Arrhythmia Database contains 48 half-hour dual-channel dynamic electrocardiogram recording excerpts from 47 subjects studied by the BIH arrhythmia laboratory between 1975 and 1979. Each record was sampled at a sampling rate of 360 Hz, with a total of 650,000 signal samples. We sliced each recorded ECG by 1024 signal samples, and 630 slices are selected for each record, so a total of 30,240 slices were used as clean ECG datasets. In the MIT-BIH Noise Stress Test Database, baseline wander noise, electrode motion artifact noise, and muscle artifact noise were obtained, which also totaled 650,000 signal samples. We also sliced each noise signal by 1024 signal samples and selected 630 slices as the noise dataset. We normalized the clean ECG dataset and noise dataset according to the following formula to ensure that the signal value of each ECG segment was between 0 and 1:

$$Normalized(x_i) = \frac{x_i - x_{min}}{x_{max} - x_{min}}, \tag{1}$$

where $x_i$ is the value of the signal in a sliced signal, $x_{max}$ is the largest value of the signal in this slice signal, and $x_{min}$ is the smallest value of the signal in this slice signal. We then used Equation (2) to add different types of noise with signal-to-noise ratios of $-5$ dB, $-2.5$ dB, 0 dB, 2.5 dB, and 5 dB to the clean ECG. These noises included Gaussian white noise,

baseline wander noise, electrode motion artifact noise, and muscle artifact noise. We used the following expressions to add the noise:

$$Y = X + N, N_i = \eta_i \sqrt{\frac{\sum x_i{}^2}{(\sum \eta_i{}^2) * 10^{\frac{snr}{10}}}} \ , \tag{2}$$

where $x_i$ is a clean ECG, $\eta_i$ is a noise signal, *snr* is the signal-to-noise ratio, and $Y$ is the ECG with noise. $N_i$ represents the noise calculated after adding a certain signal-to-noise intensity. We divided the ECG dataset into the training set, validation set, and test set in the proportion of 8:1:1. Therefore, 630 data slices of each ECG record were randomly partitioned into 504, 63, and 63 data slices according to the ratio, and a total of 24,192 data slices were used as the training set for ECG noise reduction, 3024 data slices were used as the validation set for ECG noise reduction, and 3024 data slices were used as the test set for ECG noise reduction.

*3.2. Verifying Indicators*

In our study, we used the signal-to-noise ratio (SNR), root-mean-square error (RMSE), and percentage root-mean-square error (PRD) as important indicators to measure the noise reduction effect of the ECG data. The higher the signal-to-noise ratio, the better the noise reduction effect, and the smaller the values of PRD and RMSE, the better the noise reduction effect. These three metrics can be calculated using the following three expressions:

$$RMSE = \sqrt{\sum (x_i - \hat{x}_i)^2}, \tag{3}$$

$$PRD = 100 \sqrt{\frac{\sum (x_i - \hat{x}_i)^2}{\sum x_i^2}}, \tag{4}$$

$$SNR = 10 \log_{10} \frac{\sum x_i^2}{\sum (\hat{x}_i - x_i)^2}, \tag{5}$$

where $x_i$ is the original ECG data and $\hat{x}_i$ is the ECG data after noise reduction.

**4. The Method**

ECG denoising is a basic and lasting task in ECG processing. The main challenge is to recover the clean signal ($X$) and additive noise ($N$) from the mixed signal ($Y$), namely:

$$Y = X + N, \tag{6}$$

where $Y$ represents the noisy ECG data, $X$ represents the clean ECG data, and $N$ represents the noise. This problem is ill-posed because the ECG term $X$ and the noise term $N$ are unknown and difficult to separate. Therefore, we propose a noise reduction method for separating ECG data and noise. First, we review denoising autoencoders [30] and disentangled representation learning.

*4.1. Review of Denoising Autoencoders*

Autoencoders are very popular deep learning models in the field of signal processing and image restoration. The goal of an autoencoder is to make the output data as close as possible to the input data so as to achieve the ability of data recovery. Vincent et al. proposed a denoising autoencoder based on an autoencoder, adding noise to the original input data to improve the robustness of the model. The denoising autoencoder can be divided into two stages:

(1)  Coding stage:

$$En(Y) : H = \varphi(Wy + b), \tag{7}$$

(2) Decoding stage:

$$De(H) : \hat{X} = \psi\left(\hat{W}h + \hat{b}\right), \tag{8}$$

where $En\,(\,\cdot\,)$ and $De\,(\,\cdot\,)$ represent the encoder and decoder, respectively, which are composed of a series of convolution layers and activation function layers. $Y$ represents the sum of input data ($X$) and noise ($N$), $H$ represents the feature of the hidden layer, $X$ is the output data, $W$ and $b$ are the weights and biases of the encoder, and $\hat{W}$ and $\hat{b}$ are the weights and biases of the decoder. Figure 2 shows the structure of the denoising autoencoder. The objective function of the denoising autoencoder is:

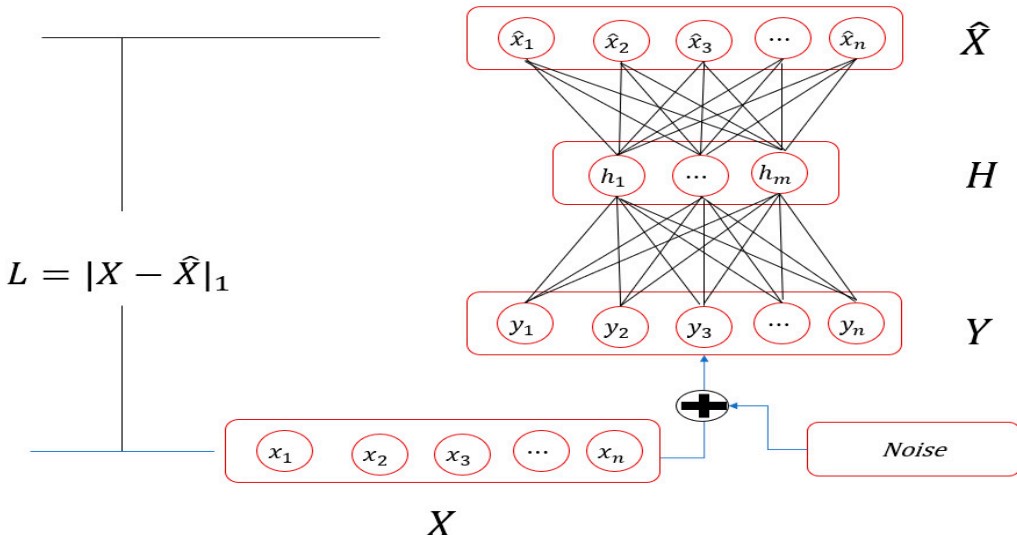

**Figure 2.** The structure of the denoising autoencoder.

*4.2. Disentangled Operator*

Disentangled representation learning is a new concept in machine learning [31]. In signal vector space, $H$ represents the coupling feature of the signal and noise, $H \in R^{C \times L}$. $H$ can be characterized as $k$ basis vectors $[h_1, h_2, \cdots, h_k]$. Each basis vector represents a more subtle feature of the signal and noise, and each subtle feature is coupled into $H$:

$$H = h_1 \otimes h_2 \otimes \cdots \otimes h_k. \tag{9}$$

Through a transformation, the coupling information of $H$ is disentangled. We define the disentangled operator:

$$f : H \to Z. \tag{10}$$

Because the characteristic basis vectors $[h_1, h_2, \cdots h_k]$ are entangled with each other, $H$ is decomposed into each decomposition factor using a disentangled transformation. They are independent of each other, which can be expressed as:

$$Z = z_1 \oplus z_2 \oplus \cdots \oplus z_j. \tag{11}$$

This is similar to the subspace direct sum of the solution space, but the difference is to solve each direct sum subspace of the transformed feature space. In the ECG denoising processing, we disentangle the noise-containing signal features using neural networks and separate them into signal features and noise features. This is to separate the feature space ($H$) of the hidden layer into two incompatible spaces ($U$ and $S$). $U$ represents the coded feature space of noise, while $S$ represents the clean ECG space.

*4.3. Proposed Method*

Using the idea of disentangled representation learning, we improve the denoising autoencoder model and propose a new autoencoder: a disentangled autoencoder. As

shown in Figure 3, unlike the denoising autoencoder, the features of the encoded hidden layer are decoupled to separate the signal-coding features and the noise-coding features. The loss function of the disentangled autoencoder can be expressed as:

$$L = L_1 + L_2 = \left|X - \hat{X}\right|_1 + \left|N - \hat{N}\right|_1. \tag{12}$$

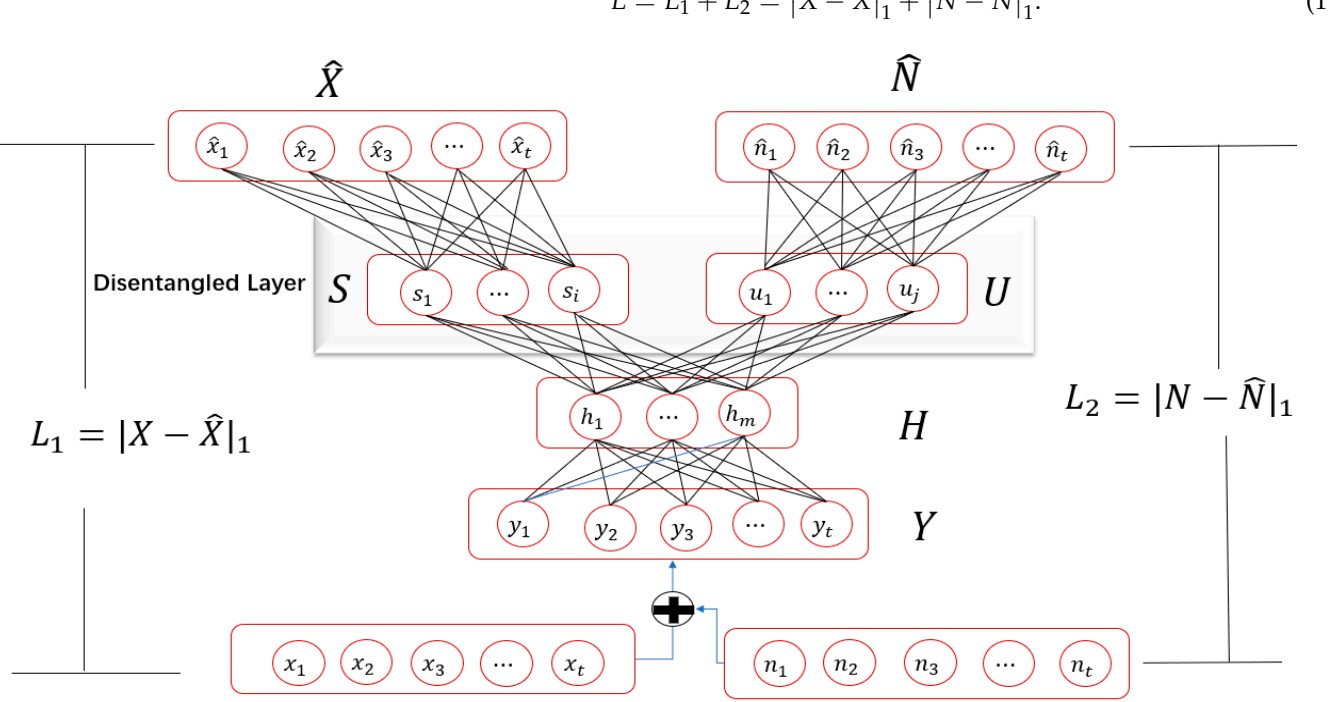

**Figure 3.** The structure of the disentangled autoencoder model.

We propose a new network structure called DANet for ECG denoising. As shown in Figure 4, the disentangled autoencoder network structure consists of three parts: an encoder, a disentangled module, and two decoders. In the encoder part, five different coding modules are used to encode the noisy input signal. In order to obtain the detailed features of the input signal, each coding module contains two convolution layers with the same convolution kernel size, a maximum pooling layer, and an empty attention module. The convolution sizes of the convolution layers of the five coding modules are $1 \times 15$, $1 \times 9$, $1 \times 7$, $1 \times 5$, and $1 \times 3$, respectively. Each pooling layer uses a convolution kernel of $1 \times 2$ and a stride size of 2 to compress the signal features. The dilated attention module (DSE block) first uses a $1 \times 1$ convolution on the input features, uses a convolution with dilated rates of 2 and 4, performs feature stacking, and finally uses a residual connection to prevent overfitting of the network model.

In the disentangled module part, this is to separate the feature map of the decoder output, the signal features, and the noise features. The noisy ECG ($Y$), with a sampling size of $1024 \times 1$, is used as the input and the output after the encoder obtains a feature map ($H$) with a channel of 512 and a size of $32 \times 1$. The feature maps ($S$ and $U$) are obtained using convolution layers with convolution kernels of $1 \times 3$ and $1 \times 5$. Then, the two feature maps are superimposed to obtain the fused feature map ($\widetilde{H}$), which can be expressed as:

$$\widetilde{H} = S + U. \tag{13}$$

Then, the feature map ($\widetilde{H}$) is input into the average pooling layer to obtain a feature map with a channel number of 1024 and a size of $1 \times 1$. Then, the attention coefficients ($\alpha$ and $\beta$) are obtained from the two fully connected layers and multiplied by the original

feature maps ($S$ and $U$) to obtain a signal feature map $\left(\widetilde{S}\right)$ and a noise feature map $\left(\widetilde{U}\right)$ with sizes of $32 \times 1 \times 1024$. The formulas can be expressed as:

$$\widetilde{S} = S \times \alpha, \tag{14}$$

$$\widetilde{U} = U \times \beta. \tag{15}$$

In the decoder part, two decoder network structures with the same structure are used to decode the signal feature map $\left(\widetilde{S}\right)$ and the noise feature map $\left(\widetilde{U}\right)$, respectively, to obtain the denoised ECG ($X$) and the approximated noise ($N$). The network structure of the decoder is similar to that of the encoder. The difference is that the maximum pooling layer in the encoder is replaced by a deconvolution layer with a convolution kernel of $1 \times 2$ and a stride size of 2. In addition, a convolution layer and a sigmoid activation function layer are used in the final output layer.

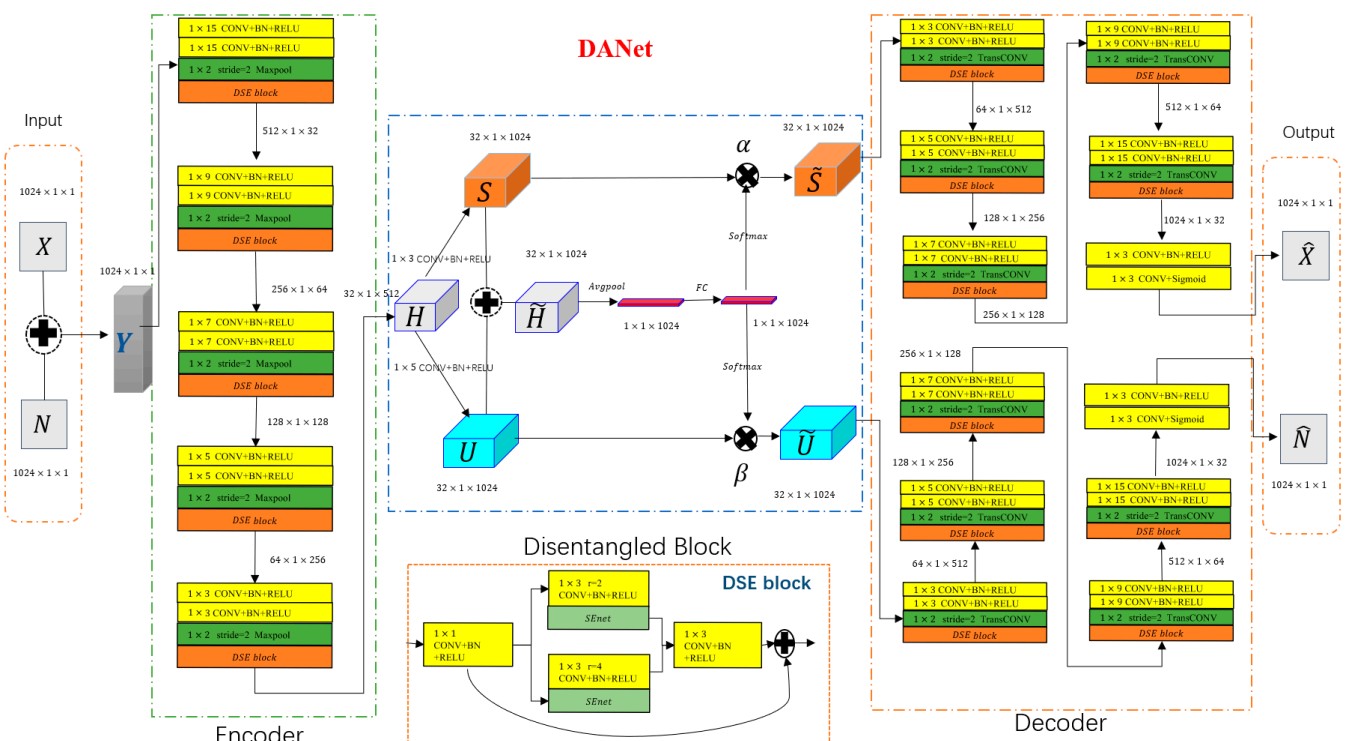

**Figure 4.** The DANet network structure.

*4.4. Model Optimization*

In our experiment, we used a GPU (NVIDIA RTX3090) with 40 G of memory. We used the python programming language and selected PyTorch as the deep learning framework for model training and testing. After repeated experiments, the batch sizes were set to 256, 128, 64, 32, and 16 at the beginning, and the learning rates of the network model training optimizer were set to 0.1, 0.05, 0.01, 0.005, 0.001, 0.0005, and 0.0001. As shown in Figure 5, the original ECG was added with Gaussian white noise with a signal-to-noise ratio of 0 dB, and the network model was trained with different batch sizes and learning rates. The experimental results show that the learning rate of our proposed network model training was 0.001 The experimental effect was the best when the batch size was set to 32. The signal-to-noise ratio of the ECG was the highest.

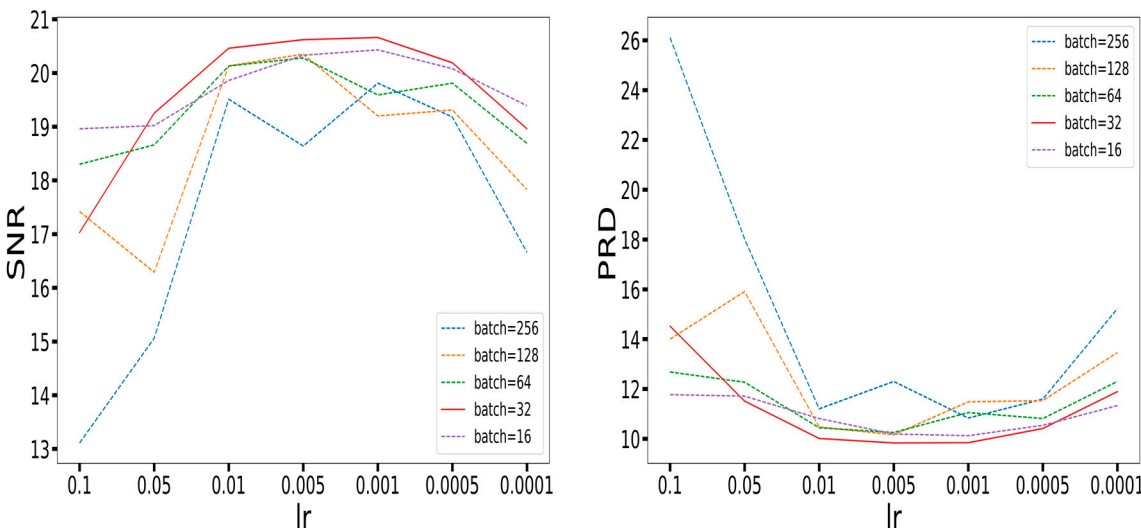

**Figure 5.** SNR and PRD of the DANet network model with different learning rates and different batch sizes.

In our experiment, we used four different optimization algorithms to compare which optimization algorithm had a better network training effect. These four optimization algorithms were stochastic gradient descent (SGD) [32], adaptive moment estimation (Adam) [33], root-mean-square prop (RMSProp) [34], and adaptive subgradient (Ada-Grad) [35]. We added 0 dB baseline wander noise to the clean ECG and then trained the proposed model with a batch size of 32 and a learning rate of 0.001 using these four different optimization algorithms. This experiment aimed to compare the performance differences of the four different optimizers under the same training settings. We used training parameters with a batch size of 32 and a learning rate of 0.001. We tested the test set to obtain the noise reduction effect of each optimizer, including the means and standard deviations of the SNR, PRD, and RMSE. The relevant data are shown in Table 1. The experimental results show that the network model training effect was better when based on Adam's optimization algorithm. The SNR reached 27.33 dB, the PRD was 4.81, and the RMSE was 0.0181. Adam's signal-to-noise ratio was about 44% higher than the SGD. As shown in Figure 6 below, our proposed network model underwent 100 iterations of training, experimental training loss, and verification loss. From the experimental results, it can be seen that the training loss and verification loss were very close, so the network model was not fitted, thus verifying the reliability of the DANet network model.

**Table 1.** The means and standard deviations of the SNR, PRD, and RMSE for the four optimizers.

| Optimizers | | Adam | SGD | RMSProp | AdaGrad |
|---|---|---|---|---|---|
| **Mean** | **SNR** | 27.33 | 15.32 | 26.62 | 22.78 |
| | **PRD** | 4.81 | 19.10 | 5.16 | 7.80 |
| | **RMSE** | 0.0181 | 0.0692 | 0.0193 | 0.0289 |
| **Std** | **SNR** | 1.40 | 2.07 | 1.523 | 1.17 |
| | **PRD** | 0.87 | 5.64 | 0.88 | 1.053 |
| | **RMSE** | 0.0052 | 0.0177 | 0.0054 | 0.0066 |

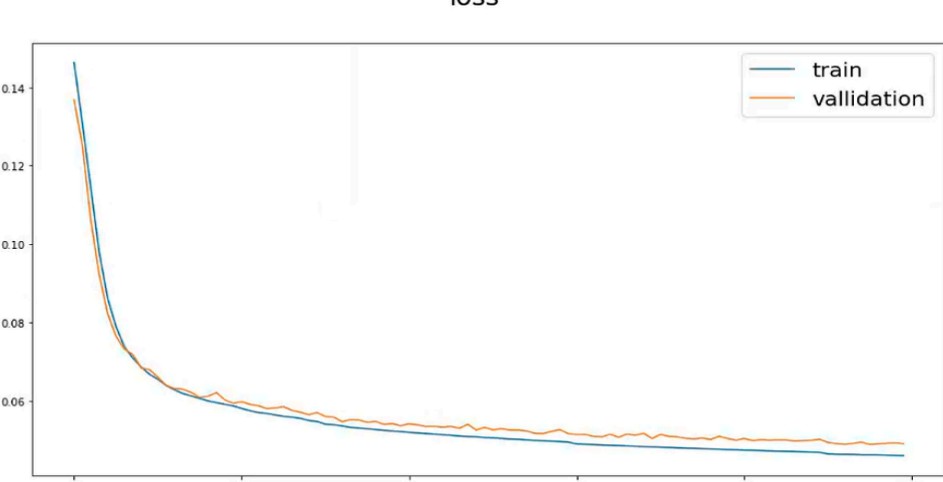

**Figure 6.** Loss of training set and loss of validation set.

## 5. The Experiment

### 5.1. Denoising Performance of the Proposed Model

To verify the robustness of our proposed network model to different types of ECG noises, we randomly selected different numbers of signals from the MIT-BIH Arrhythmia Database and then added four different forms of ECG noises to these signals, specifically Gaussian white noise, baseline wander noise, electrode motion artifact noise, and muscle artifact noise. Gaussian white noise was simulated by generating random noise, while baseline wander noise, electrode motion artifact noise, and muscle artifact noise were simulated by the three corresponding types of noise in the MIT-BIH Noise Stress Test Database. Figure 7 shows the noise reduction effect of the 220th recorded ECG in the MIT-BIH Arrhythmia Database after adding baseline wander noises at an input SNR of 5 dB. We can see that the original ECG had baseline wander after adding the baseline wander noise, which was not conducive to a doctor's diagnosis. In the denoising process, our proposed method effectively restored the baseline characteristics of the original signal. Figure 8 shows the noise reduction effect of the 123rd ECG recorded in the MIT-BIH Arrhythmia Database after adding electrode motion noise at an input SNR of 0 db. It can be seen in Figure 8 that the original ECG was severely damaged after adding the noise. The P wave and T wave disappeared completely. The QRS wave still existed, but some deformation also occurred. In the denoising process, the signal details of the P wave and T wave were reproduced, and the QRS band was completely restored.

Figure 9 shows the noise reduction effect diagram of the 117th recorded ECG in the MIT-BIH Arrhythmia Database after adding muscle artifact noise at an input SNR of 2.5 db. After adding muscle artifact noise to the original ECG, the original ECG retained the signal characteristics of the R wave. In addition, other bands were damaged by different degrees of waves. Most of the signal waves were changed into wave shapes, and the original P wave, T wave, Q wave, and other characteristic band information were lost. In the process of denoising, not only was all the characteristic wave information restored, but the ECG was also smoother. Figure 10 shows the noise reduction effect diagram of the ECG recorded in the MIT-BIH Arrhythmia Database after adding Gaussian white noise at an input SNR of 5 db to the ECG recorded in the MIT-BIH Arrhythmia Database. It can be seen in Figure 10 that after adding Gaussian white noise at an input SNR of 5 db, the characteristic information of each band of the original ECG was completely lost. Compared with the three other types of noise, the wave damage of Gaussian white noise was the most thorough, and it was impossible to extract useful information. This damage was catastrophic to a doctor's diagnosis. In the process of ECG denoising, our proposed method could make the denoised ECG retain the morphological characteristics of each band and also made the ECG smoother. Even after adding Gaussian white noise, the band features of

the original ECG were completely lost, and the proposed method could still recover the feature information of each band of the ECG. This verifies that our method can effectively eliminate Gaussian white noise.

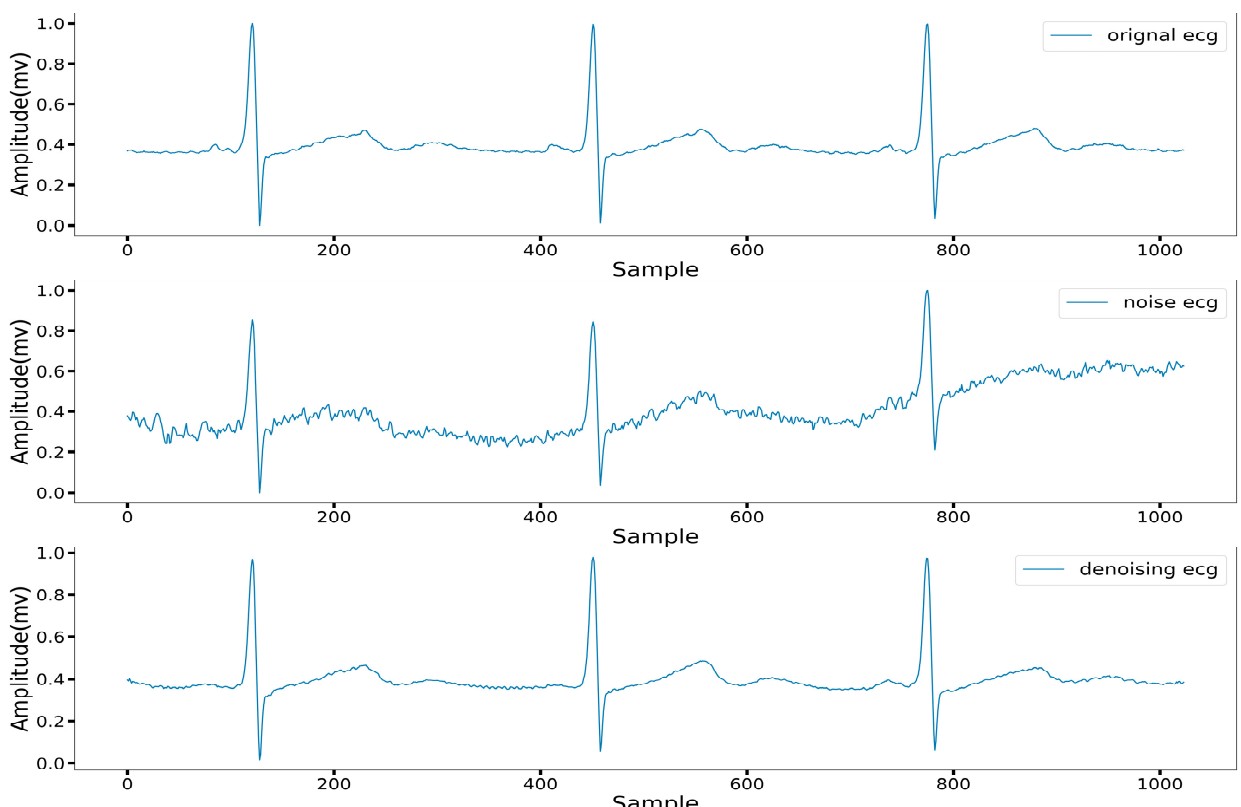

**Figure 7.** Noise reduction effect of ECG with baseline wander noise at input SNR of 5 db.

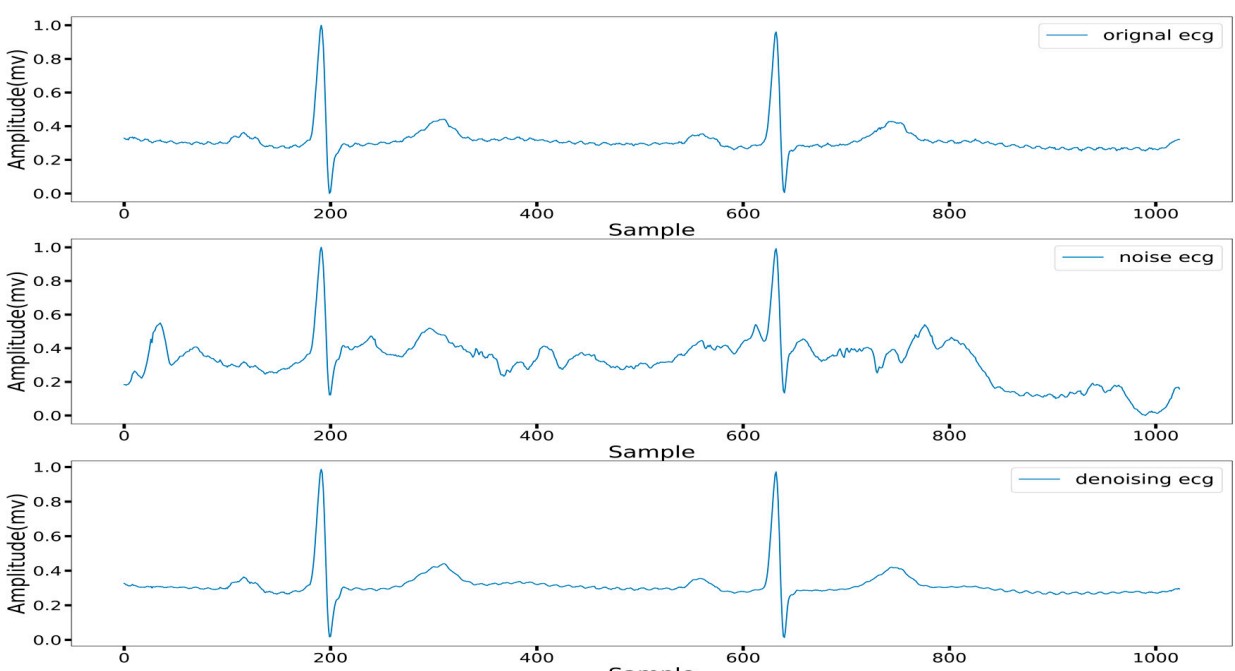

**Figure 8.** Noise reduction effect of ECG with electrode motion artifact noise at input SNR of 0 db.

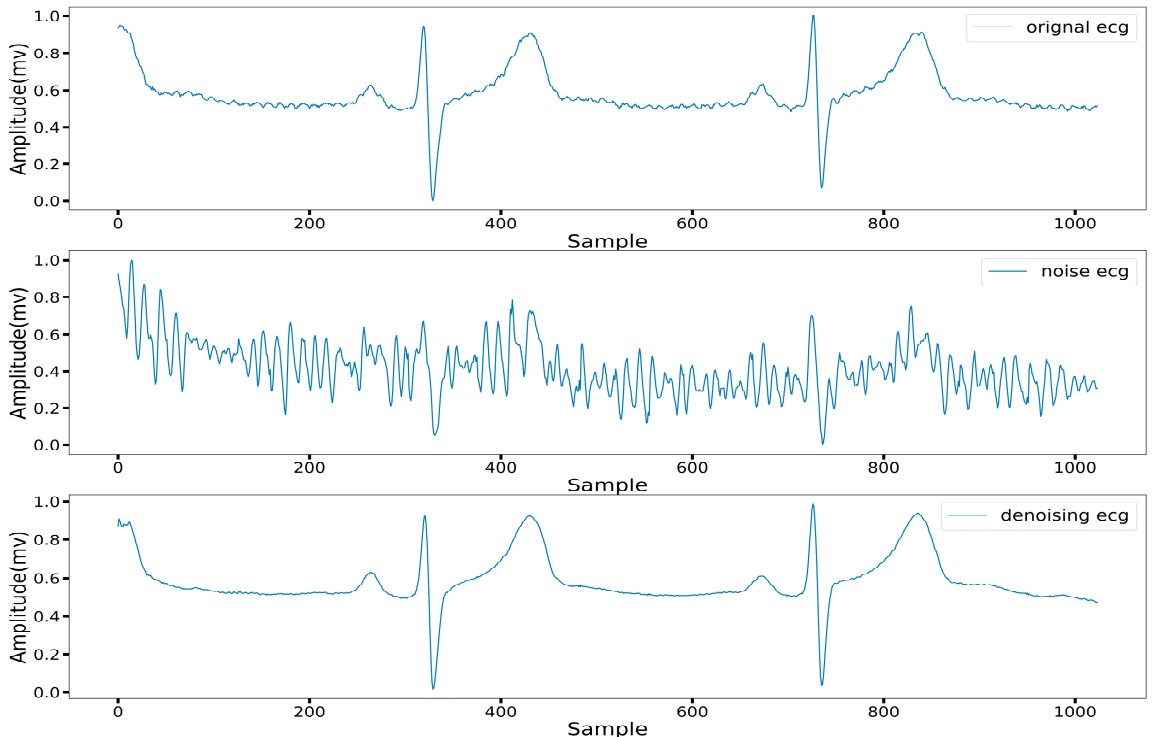

**Figure 9.** Noise reduction effect of ECG with muscle artifact noise at input SNR of 2.5 db.

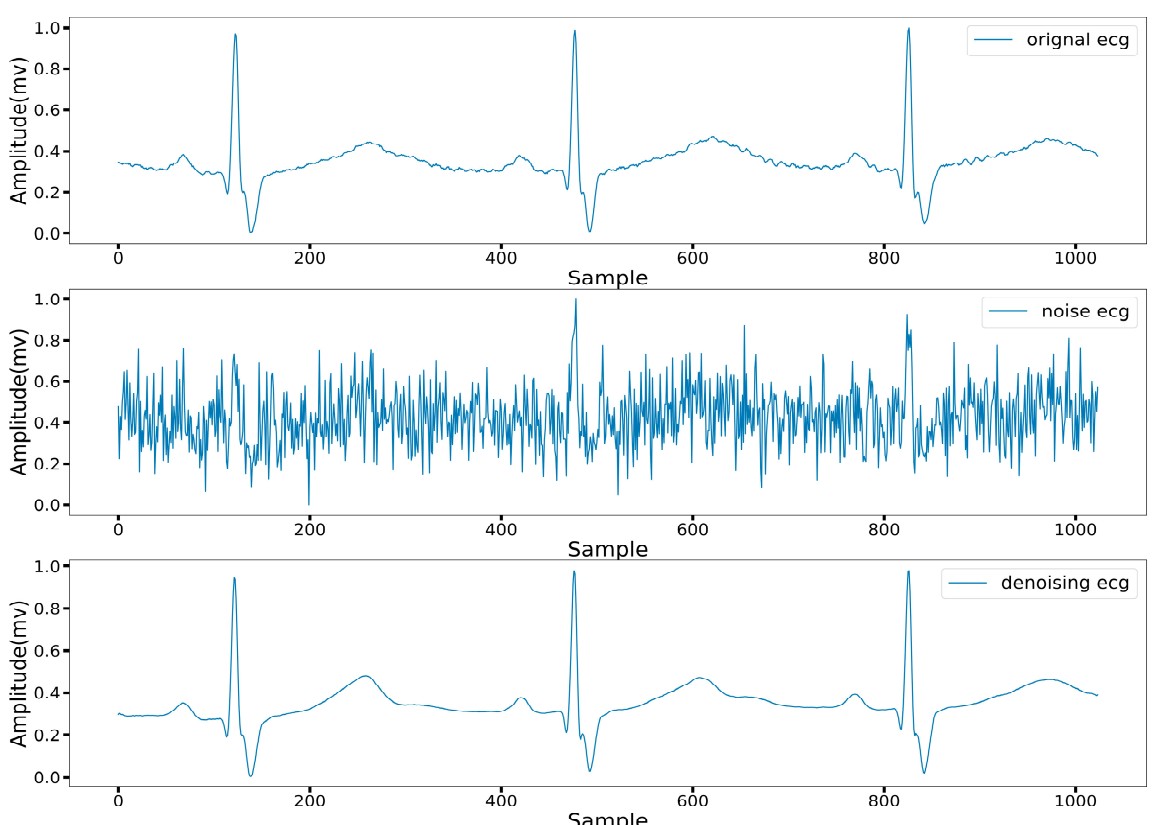

**Figure 10.** Noise reduction effect of ECG with Gaussian white noise at input SNR of 5 db.

*5.2. The Contrast Experiment*

To compare the denoising performance of our proposed method, we chose an autoencoder based on a deep fully connected neural network (hereafter referred to as DNN) [36]

and a denoising autoencoder based on a fully convolutional neural network (hereafter referred to as FCN) [30] as the comparison objects. To make the comparison experiments more scientific, we set the DNN, FCN, and DANet all to train on the same training set and test on the same test set. For fairness, we used Adam's optimization algorithm to train all the models. The batch size of the training dataset was set to 32, the learning rate was 0.001, the loss function was the root-mean-square error between the original ECG and the denoised ECG, and the number of training iterations was set to 100. In addition, all models were batch-normalized, and the activation function was the RELU function. At the 0 dB input signal-to-noise ratio noise level, Tables 2–4 represent a quantitative comparison of the denoising effects of the various autoencoder models after adding baseline wander noise, muscle artifact noise, and electrode motion artifact noise, respectively. In these three tables, it can be seen that using different ECG noise reduction performance indicators, the proposed denoising framework based on a disentangled autoencoder was superior to a fully convolutional denoising autoencoder and a fully connected autoencoder. These quantitative performance evaluations demonstrate the strength of the proposed disentangled-autoencoder-based denoising framework. The proposed framework has good generalization for EM, MA, and BW noises. It should be noted that it can be further enhanced by larger training datasets and better DANet model design. In addition, multiple noise cases can be trained simultaneously using a single DANet model. In the future, we hope to use different databases and other noise conditions to train the model.

**Table 2.** Quantitative comparison of denoising effects of various autoencoder models after adding baseline wander noise at 0 dB input SNR noise level.

| Record | DANet | | | FCN | | | DNN | | |
|---|---|---|---|---|---|---|---|---|---|
| | SNR | RMSE | PRD | SNR | RMSE | PRD | SNR | RMSE | PRD |
| **109** | 26.82 | 0.0196 | 5.24 | 25.1 | 0.0236 | 6.31 | 17.02 | 0.0526 | 14.5 |
| **117** | 27.7 | 0.0215 | 4.31 | 25.92 | 0.0265 | 5.29 | 17.92 | 0.0644 | 12.88 |
| **118** | 26.23 | 0.0249 | 5.29 | 24.74 | 0.0293 | 6.23 | 17.16 | 0.066 | 14.04 |
| **119** | 26.5 | 0.0156 | 5.19 | 24.44 | 0.0194 | 6.38 | 15.23 | 0.0538 | 18.34 |
| **123** | 29.19 | 0.0127 | 3.78 | 27.24 | 0.0158 | 4.71 | 16.45 | 0.0514 | 15.37 |
| **215** | 26.76 | 0.0207 | 4.87 | 24.73 | 0.0255 | 6.14 | 15.1 | 0.075 | 17.68 |
| **220** | 30.63 | 0.0114 | 3.08 | 28.94 | 0.0137 | 3.74 | 16.15 | 0.058 | 15.71 |
| **230** | 26.55 | 0.0212 | 5.24 | 24.84 | 0.0252 | 6.19 | 16.46 | 0.0627 | 15.21 |
| **231** | 27.17 | 0.0141 | 4.9 | 25.78 | 0.0163 | 5.7 | 14.31 | 0.0591 | 20.48 |
| **233** | 26.96 | 0.0252 | 5.13 | 24.8 | 0.032 | 6.58 | 18.57 | 0.0597 | 12.18 |
| **Average** | **27.451** | **0.01869** | **4.703** | **25.653** | **0.02273** | **5.727** | **16.437** | **0.06027** | **15.639** |

**Table 3.** Quantitative comparison of denoising effects of various autoencoder models after adding muscle artifact noise at 0 dB input SNR noise level.

| Record | DANet | | | FCN | | | DNN | | |
|---|---|---|---|---|---|---|---|---|---|
| | SNR | RMSE | PRD | SNR | RMSE | PRD | SNR | RMSE | PRD |
| **109** | 26.15 | 0.0182 | 5.09 | 25.28 | 0.02 | 5.59 | 16.89 | 0.051 | 14.43 |
| **117** | 25.48 | 0.028 | 5.59 | 23.6 | 0.0345 | 6.88 | 17.13 | 0.0706 | 14.07 |
| **118** | 24.99 | 0.028 | 5.96 | 23.58 | 0.0324 | 6.89 | 16.71 | 0.0693 | 14.72 |
| **119** | 24.52 | 0.0187 | 6.31 | 23.14 | 0.0217 | 7.35 | 14.75 | 0.0552 | 19.46 |
| **123** | 27.5 | 0.0144 | 4.35 | 25.95 | 0.0173 | 5.21 | 16.29 | 0.0512 | 15.49 |
| **215** | 24.74 | 0.0249 | 6.04 | 23.02 | 0.0303 | 7.4 | 14.77 | 0.0776 | 18.33 |
| **220** | 28.41 | 0.0146 | 3.97 | 26.32 | 0.0184 | 5.02 | 15.8 | 0.0602 | 16.35 |
| **230** | 25.05 | 0.0259 | 6.01 | 23.51 | 0.0303 | 7.01 | 16.4 | 0.0663 | 15.26 |
| **231** | 25.08 | 0.0165 | 5.9 | 23.2 | 0.0202 | 7.39 | 13.62 | 0.059 | 22.16 |
| **233** | 25.29 | 0.028 | 5.89 | 23.16 | 0.0354 | 7.42 | 17.64 | 0.0649 | 13.56 |
| **Average** | **25.721** | **0.02172** | **5.511** | **24.076** | **0.02605** | **6.616** | **16** | **0.06253** | **16.383** |

**Table 4.** Quantitative comparison of denoising effects of various autoencoder models after adding electrode motion artifact noise at 0 dB input SNR noise level.

| Record | DANet | | | FCN | | | DNN | | |
|---|---|---|---|---|---|---|---|---|---|
| | SNR | RMSE | PRD | SNR | RMSE | PRD | SNR | RMSE | PRD |
| 109 | 28.54 | 0.0139 | 3.95 | 23.5 | 0.0245 | 6.95 | 14.14 | 0.0693 | 19.82 |
| 117 | 31.03 | 0.0145 | 2.87 | 24.12 | 0.0325 | 6.39 | 14.83 | 0.0927 | 18.33 |
| 118 | 29.02 | 0.0177 | 3.71 | 23.66 | 0.0326 | 6.88 | 13.42 | 0.1017 | 21.44 |
| 119 | 29 | 0.0111 | 3.76 | 22.33 | 0.0238 | 8.13 | 11.56 | 0.0796 | 27.88 |
| 123 | 31.98 | 0.0098 | 2.85 | 26.4 | 0.0181 | 5.28 | 14.31 | 0.0662 | 19.56 |
| 215 | 28.6 | 0.0168 | 3.87 | 23.72 | 0.0295 | 6.86 | 11.75 | 0.1143 | 25.92 |
| 220 | 32.92 | 0.0087 | 2.31 | 28.07 | 0.0153 | 4.08 | 14.17 | 0.0738 | 19.7 |
| 230 | 30.13 | 0.0137 | 3.3 | 23.69 | 0.0285 | 6.83 | 14.1 | 0.0837 | 19.91 |
| 231 | 28.75 | 0.0108 | 3.96 | 22.99 | 0.0211 | 7.46 | 11.45 | 0.0784 | 27.83 |
| 233 | 29.18 | 0.0185 | 3.66 | 23.05 | 0.0375 | 7.5 | 14.1 | 0.1013 | 20.02 |
| Average | 29.915 | 0.01355 | 3.424 | 24.153 | 0.02634 | 6.636 | 13.383 | 0.0861 | 22.041 |

## 6. Discussion

As a model for ECG denoising, DANet is a new idea based on a disentangled autoencoder, which has certain advantages and disadvantages. The disentangled autoencoder used in DANet can effectively reduce the noise interference of ECG and improve the signal quality. The model was designed based on deep learning technology, which can use large-scale data for training to improve prediction ability and accuracy. However, DANet also has some limitations and shortcomings. Firstly, due to the complex structure of the decoupled autoencoder, a large amount of computing resources and time are required during training. At the same time, the model needs a certain amount of time to process during testing, and there may be a certain delay. In terms of training strategies, it is necessary to consider the issue of protecting patient privacy and how to train and test efficiently on limited datasets. In our experimental design, the lack of separation of patient data may have affected our accurate assessment of model performance. Therefore, in order to more accurately evaluate the performance of the model, we have taken some measures. Specifically, we separated the data according to the patient record number. We selected six ECG data with record numbers of 107, 115, 123, 207, 220, and 233 as the test set and selected the remaining patient record data as the training set. The ratio of the training set and test set was 7:1. In this way, we could better protect the privacy and security of the patient data, more accurately evaluate the performance of the model, and improve the credibility of the experimental design. According to the experimental results in Table 5, compared with the experiment (such as in Tables 2–4) without patient data separation, the noise reduction effect of the baseline wander noise of the DANet model was reduced by 11.47%, and the noise reduction effect of the electrode motion artifacts was reduced by 5.36%, while the signal-to-noise ratio of the DANet model increased by 7.46% when dealing with muscle artifact noise. The experimental results show that although the signal-to-noise ratio of the DANet model decreased slightly in the patient data separation experiment, it could still maintain a good noise reduction effect. Therefore, it can be concluded that the noise reduction ability of the DANet model still performed well in different types of noise processing after the separation of patient data.

**Table 5.** The noise reduction effect of the DANet model using a training set and a test set divided according to different patient data.

| Record | BW | | | EM | | | MA | | |
|---|---|---|---|---|---|---|---|---|---|
| | SNR | RMSE | PRD | SNR | RMSE | PRD | SNR | RMSE | PRD |
| **107** | 26.05 | 0.0279 | 5.51 | 27.75 | 0.0219 | 4.30 | 28.18 | 0.0202 | 3.96 |
| **115** | 26.62 | 0.0194 | 5.10 | 28.98 | 0.0139 | 3.68 | 26.93 | 0.0175 | 4.66 |
| **123** | 19.77 | 0.0398 | 10.65 | 25.59 | 0.0215 | 5.78 | 24.81 | 0.0220 | 5.90 |
| **207** | 22.03 | 0.0466 | 8.58 | 28.73 | 0.0217 | 3.99 | 29.79 | 0.0179 | 3.28 |
| **220** | 25.20 | 0.0219 | 5.90 | 28.51 | 0.0143 | 3.86 | 26.18 | 0.0186 | 5.03 |
| **233** | 26.11 | 0.0286 | 5.41 | 30.31 | 0.0165 | 3.13 | 29.96 | 0.0169 | 3.20 |
| **Average** | **24.30** | **0.0307** | **6.86** | **28.31** | **0.0183** | **4.12** | **27.64** | **0.0189** | **4.34** |

## 7. Conclusions

We introduced a disentangled mechanism module into a traditional autoencoder and proposed a new autoencoder model. We used disentangled representation learning to decouple the hidden layer features after coding, achieved the separation of signal and noise features, and then obtained the denoised signal by decoding. In this paper, different types of noise were added, which proved that our proposed denoising method has good robustness. Meanwhile, the effectiveness of the proposed method was evaluated using the signal-to-noise ratio, percentage root-mean-square error, and mean square error. In experiments, compared with models such as an FCN-based denoising autoencoder and a DNN-based autoencoder, our proposed method had more advantages in the signal-to-noise ratio and PRD. In future work, we will explore how to better extract the features of ECG, separate ECG features and noise features, improve the noise reduction effect of ECG, and better apply these findings to an automatic diagnosis system of ECG.

**Author Contributions:** Conceptualization, H.L., R.L. and Z.L.; methodology, H.L., R.L. and Z.L.; software, H.L.; validation, H.L., R.L. and Z.L.; formal analysis, H.L., R.L. and Z.L.; investigation, H.L., R.L. and Z.L.; resources, H.L.; data curation, H.L.; writing—original draft preparation, H.L.; writing—review and editing, H.L., R.L. and Z.L.; visualization, H.L.; supervision, R.L. and Z.L.; project administration, R.L. and Z.L.; funding acquisition, R.L. and Z.L.; All authors have read and agreed to the published version of the manuscript.

**Funding:** This work is supported by the National Natural Science Foundation of China under Grant No. 62172243, the Natural Science Foundation of Shandong Province under Grant No. ZR2021MF084 and ZR2020QF020.

**Data Availability Statement:** The data presented in this study are available upon request from the corresponding author.

**Conflicts of Interest:** The authors declare that they have no conflict of interest.

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
