# Peer review of "ECG Signal Denoising Method Based on Disentangled Autoencoder"

_electronics, doi:10.3390/electronics12071606_

Round 1

Reviewer 1 Report

The manuscript presents a neural network for ECG denoising based on a convolutional encoder, a disentangled module and two convolutional decoders, named disentangled autoencoder (DANet). The methodology is of scientific interest, however, the manuscript should be considerably improved in order to become suitable for publication in a high-quality journal. Here below are some questions, remarks and recommendations that could help the authors in the revision of the paper:

1)      The abstract does not present any useful information about the study. It should contain brief description of the used databases, applied methods and achieved results.

2)      Revise the sentence “Muscle artifacts is a high-frequency noise, the frequency is generally between 30-300Hz, is due to the body ' s muscles caused by tremors noise interference.”

3)      The aim of the study should be clearly stated at the end of the Introduction.

4)      The quality of all figures should be considerably improved, since they are not readable in their present form.

5)      I recommend the authors to follow the standard article structure and to define the standard sections – Database description, Methods, Results, Discussion.

6)      The authors should describe very carefully the used ECG and noise signals. They address “MIT-BIH pressure noise database”, “motor interference noise”, “electrode interference noise” that are not consistent with the content of the MIT database. The database contains baseline wander (in record 'bw'), muscle (EMG) artifact (in record 'ma'), and electrode motion artifact (in record 'em').

7)      Recommendation: Substitute “sampling point value” with “ECG value” or “signal value”.

8)      Rewrite the following sentence: “Then, we add the specified signal-to-noise ratio noise to the clean ECG signal data.” Where is SNR specified?

9)      Check the equation for Ni (12). If it is correct, it needs an explanation.

10)  The authors should explain what is the content of the training, validation and test set (in section Database description). Is the separation of data done patient-wise (i.e. ECG segments from one and the same patient should not be present in more than one set (training OR validation OR test). Otherwise, the test results are biased, since they are not obtained on an independent ECG signals.

11)  Subsection “Experimental parameters” should be part of section ‘Methods’. It describes the optimization of the model, so I recommend the authors to rename it appropriately – e.g. Model optimization. The description of the optimization process should be improved. How is it done – when optimizing the batch size what happens with the learning rate and the optimization algorithm (are they constant) and vice versa? Ate all values optimized at one and the same time randomly, or the authors have followed some logic (e.g. first LR is optimized for presser batch size and optimizer, than ….)? Describe the applied optimization procedure.

12)  What are the values for SNR, PRD and RMSE in Table 1? Are these the measured mean values for the training, or for the test database, or something else? If these are mean values – provide also the standard deviation.

13)  In subsection ‘Denoising Performance of the Proposed Model’:

-          What do the authors mean when they wrote “…adding 5dB baseline wander noise”, “after adding 0dB electrode interference noise”, “adding 2.5dB muscle artifacts noise”, etc.? The noises are added with some amplitude or at predefined SNR.

-          “In the denoising process, our proposed method restores the baseline characteristics of the original signal well.” and also “The P wave and T wave disappear completely, and the QRS wave still exists, but some deformation also occurs.” These are subjective estimations. Provide some numerical assessment – i.e SNR, RMSE, PRD at the input and at the output of the model.

-          “electrode interference noise” – there is not such noise type in MIT database. Correct such errors everywhere in the manuscript!

14)  Are the DNN and FCN (used for comparison) trained and tested on the same datasets as DANet? This should be clearly written in the manuscript.

15)  Section ‘Discussion’ is missing. It should contain discussion about the positive and negative sides of the proposed DANet, as well as some discussion on the training strategy (patient-wise or not) and limitations related to this strategy (if the test is not independent).

16)  The last but not the least – the language should be considerably improved. If this is not done, the quality of the manuscript would be compromised and it will not be suitable for publication.

Author Response

Dear Reviewer 1,

  We have tried our best to revise the manuscript according to youer comments. Thank you very much.

Reviewer 2 Report

The paper needs to be improved:

results and discussion

Author Response

We have tried our best to revise the manuscript according to your comments. Thank you very much.

Round 2

Reviewer 1 Report

The authors have considered part of the recommendations in my first review. However, some of the points they have addressed only in their answers to the reviewer, but no comments and data are added to the text of the manuscript. All such points should be incorporated in the text in a different color in order to be visible. Here below is a list of points that are not currently addressed in the manuscript:

1)      The quality of the figures is still compromised. The numbers on the x,y-axes in Fig. 5,7,8,9,10 are not visible. Some of the horizontal lines that wrap the subplots are also missing.

2)      The authors still address “MIT-BIH pressure noise database” which is not the correct name of the database.

3)      The authors have made some inadequate corrections, i.e. “650,000 signal values”, “1024 signal value”, etc. In these cases the correct is “signal samples”. And also “?? is the value of a signal value in a sliced signal” – this should be “?? is the value of the signal in a sliced signal” or “?? is the signal value in a sliced signal”. The authors should do the corrections carefully and if possible with the help of a person who is familiar with the terminology.

4)      The authors have written that “630 data slices of each ECG signal record were randomly partitioned into 504, 63 and 63 data slices according to the ratio, and a total of 24192 data slices were used as the training set for ECG signal noise reduction, 3024 data slices as the validation set for ECG signal noise reduction and 3024 data slices as the test set for ECG signal noise reduction.” This means that they have not separated the data patient-wise and this should be pointed out as a limitation of the study (either in section Discussion or in a separate section Limitation). The authors should comment this in the aspect that the test set is not independent and the reported results could be biased towards higher values.

5)      In the caption of Table 1 the authors should specify that these are mean values +/- standard deviation, and should also include the values of the std in the table.

6)      Section Discussion should be extended.

7)      Despite the authors have made some changes based on some reviewers suggestions, the manuscript still needs considerable language improvement. If this is not done, the quality of the manuscript would be compromised and it will not be suitable for publication.

Author Response

Dear Reviewer,

   We have revise the manuscript according to your comments. Thank you very much.

Round 3

Reviewer 1 Report

The authors have addressed all my recommendations and now the content of the manuscript is suitable for publication. I hope that the necessary language corrections would be made by the Editorial team of the journal.